# *MIP-Bench*: CAN LLMS IMPLICITLY PERSONALIZE RESPONSES USING LONG-TERM MEMORY?

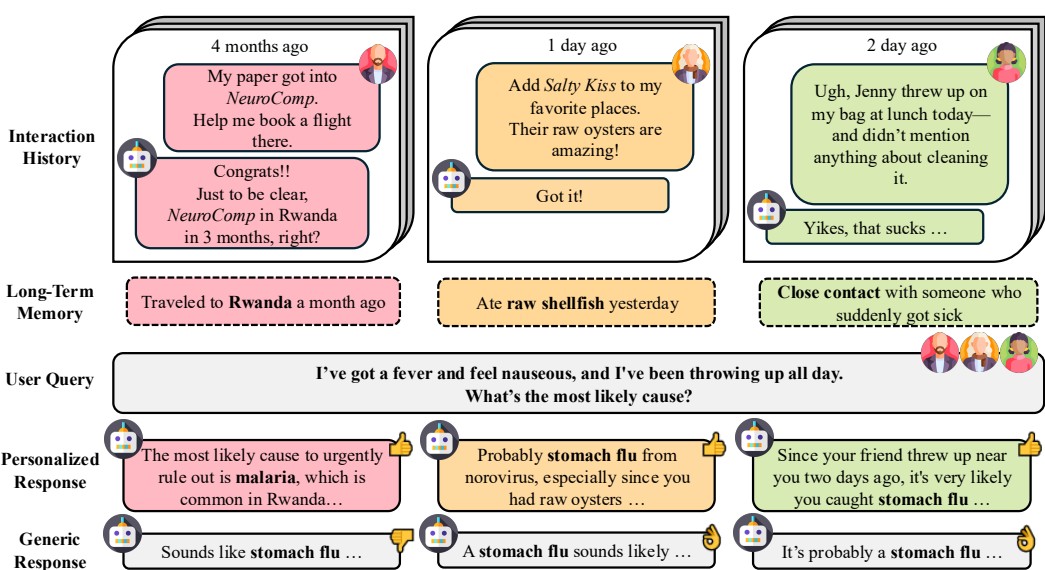

Figure 1: An example from our *MIP-Bench* illustrating how queries of an user yield distinct responses when models implicitly incorporate personal context. Responses grounded in user-specific information—such as recent travel history, dietary intake, or social exposure—demonstrate greater relevance and specificity compared to generic outputs.

## ABSTRACT

Implicit Personalization (IP) is the task of tailoring responses to individual users by implicitly inferring their personal context. Prior studies in IP typically infer context from a single prompt. However, as interactions accumulate, users expect **Memory-driven Implicit Personalization (MIP)**, where models implicitly leverage contexts from users' long-term interaction histories to provide more helpful responses. MIP introduces two unique challenges: (i) identifying sparse yet relevant personal contexts within extensive historical interactions, and (ii) understanding how varying personal contexts influence preferences among plausible answers to differentiate responses between multiple users. To navigate these challenges, we introduce *MIP-Bench*, the first benchmark to evaluate MIP in large language models (LLMs). 514 queries are assessed across an average of 6.4 users, each having approximately 196 interaction histories. Our experiments reveal that recent LLMs struggle with MIP, primarily due to difficulties in identifying and retrieving relevant personal context from memory. Furthermore, our new distribution-level evaluation framework shows that even models with strong instance-level performance often fail to differentiate responses across users, defaulting to generic or overly broad outputs rather than personalized ones.

# 1 INTRODUCTION

Personalization in large language models (LLMs) aims to generate responses tailored to individual user contexts (Zhang et al., 2024b). A response can be regarded as personalized if it is preferred by a user due to their particular personal context. Early research (Afzoon et al., 2024; Santurkar et al., 2023; Durmus et al., 2023) primarily explored **Explicit Personalization**, where personal context is directly included in the user's query—for instance, by prepending a phrase such as *"Generate a response suitable for a PhD student in biology."* While intuitive, requiring users to repeatedly specify their desires in the prompt is far from a natural interaction, and furthermore, there are cases where a user's intent is so abstract that it is difficult to articulate verbally.

To address these limitations, recent research (Peng et al., 2025; Jin et al., 2024) has begun exploring **Implicit Personalization**, where LLMs infer what the user expects from previous interactions, rather than expecting them to explicitly state it. However, prior work has largely reduced implicit personalization to the task of extracting context from isolated, single-turn cues—for example, distinguishing American versus British English through word choice (*color vs. colour*) (Jin et al., 2024). This limited perspective overlooks the potential of **Memory-driven Implicit Personalization** (**MIP**), where LLMs implicitly leverage rich personal contexts accumulated from a history of interactions to effectively personalize responses. Consider the query in Figure 1: *"I've got a fever, feel nauseous, and have been throwing up all day. What's the most likely cause?"* The query alone lacks sufficient cues for personalization. Yet, historical interactions can provide critical context: User 1's recent travel to Rwanda could suggest malaria, whereas User 2's recent eating of raw shellfish may point toward stomach flu.

We introduce *MIP-Bench*, a novel dataset and evaluation protocol for memory-driven implicit personalization in LLMs. *MIP-Bench* comprises 379 users with extensive conversational histories with an AI assistant. Each of the 514 queries is presented to an average of 6.4 users, spanning domains from expert domains (e.g., legal, medical) to casual domains. The queries are designed so that different users yield distinct answers grounded in their prior interactions and personal contexts. To evaluate model outputs, we provide query-specific rubrics that convert open-ended responses into standardized representations. Building on this foundation, we propose two complementary metrics: the *Instance Personalization Score (IPS)*, which measures how accurately a model tailors a response to an individual user's context, and the *Distribution Personalization Score (DPS)*, which assesses whether responses are sufficiently diverse across users with different contexts. Together, these metrics establish a rigorous framework for evaluating memory-driven implicit personalization.

Experiments on *MIP-Bench* reveal critical shortcomings of current LLMs on Memory-driven Implicit Personalization. First, the dominant bottleneck is not downstream response generation, but the upstream task of implicitly locating and prioritizing the right personal signals within long interaction histories. Unlike traditional settings that provide the relevant profile or traits and then ask the model to align with them, *MIP-Bench* requires a reverse workflow: start from the query, hypothesize which aspects of personal context could matter, and then navigate the interaction history to retrieve, verify, and select the decisive evidence. Second, we observe two recurring failure modes of LLMs, as illustrated in Figure 1. (i) Majority guessing—defaulting to the statistically frequent answer (e.g., "stomach flu") even when user-specific evidence supports a minority outcome; and (ii) Catch-all responses—hedging by enumerating all plausible options rather than committing to a narrow, personalized one. Finally, models that score highly at the instance level (e.g., GPT-4o (OpenAI, 2024) and Gemini-1.5-Pro (Team et al., 2024)) still underperform on the distribution metric, indicating limited sensitivity to cross-user variation despite strong general reasoning.

# 2 RELATED WORK

**Memory Systems.** Human memory is often split into declarative and non-declarative (Squire, 1992). Declarative memory includes facts and events that can be consciously recalled and verbalized. Non-declarative memory includes skills, habits, and conditioned responses that are expressed through performance rather than explicit recall. We use this distinction to organize related work.

**Long-term Memory Benchmarks.** Long-term memory benchmarks address the question: *"Can the model remember C?"*, where $C$ denotes a unit of personal context (Zhang et al., 2024a). They

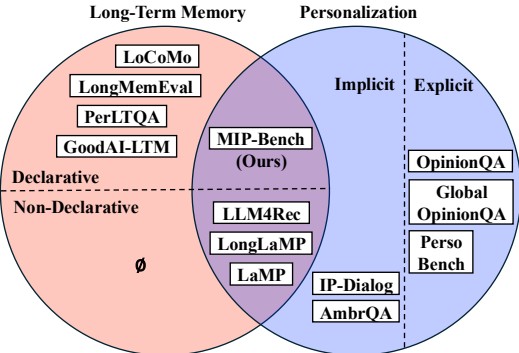

Figure 2: Venn diagram showing the position of *MIP-Bench* relative to existing personalization and long-term memory benchmarks, highlighting its unique position at the intersection of declarative memory and implicit personalization.

investigate how to store, retrieve, and update information across extended interaction histories. Representative suites such as LoCoMo (Maharana et al., 2024), LongMemEval (Wu et al., 2024), and PerLTQA (Du et al., 2024) provide multi-session conversational traces and tasks (e.g., QA, summarization) to assess these memory operations. While these works probe memory competence, they do not require personalize the final response to the user's current needs beyond recalling $C$.

**Memoryless Personalization Benchmarks.** Personalization benchmarks (Zhang et al., 2024b) ask the question: *"Can we the model its response to a specific user?"* Prior work (Afzoon et al., 2024; Santurkar et al., 2023; Durmus et al., 2023) has largely focused on *explicit personalization*: given a specific context $C$, generate a response aligned with that perspective. However, assuming $C$ is supplied in the prompt is either unrealistic or burdensome for users. This has motivated *implicit personalization*, where the system infers $C$ and adapts its response accordingly. AmbrQA (Jin et al., 2024) infers American vs. British English, socioeconomic background, or misinformation stance from subtle linguistic cues in the user query (e.g., color vs. colour). IP-Dialog (Peng et al., 2025) incorporates a short prepended conversation snippet as the source of twelve predefined user attributes. However, inference is one-shot; the model infers $C$ from a single user prompt. Furthermore, user attributes span a fixed schema, constraining what can be discovered and justified. Users expect implicit personalization to utilize long-term memory—a mechanism that incrementally builds an open-ended user model across extended interactions and grounds responses.

**Memory-driven Personalization Benchmarks.** In recommendation settings like LLM4Rec (Shehmir & Kashef, 2025), datasets such as MovieLens (Harper & Konstan, 2015) or Amazon Product Reviews (Ni et al., 2019) support personalized prediction of next items based on a user history. LaMP (Salemi et al., 2023) and LongLaMP (Kumar et al., 2024) broaden this paradigm with stylistic imitation tasks (e.g., paraphrasing tweets in a user's style). PersonalLLM (Zollo et al., 2024) proposes using RLHF reward models to simulate diverse user preferences. Personalization is typically framed as a sequence prediction over task logs, teaching models latent regularities such as habits or stylistic tendencies from behavioral data. Such capability is analogous to human non-declarative memory like the information that we remember and use unconsciously.

In contrast, *MIP-Bench* targets personalization grounded in declarative memory; conscious, intentional recollection of factual information, previous experiences, and concepts, asking *"Can the model tailor its response to a specific user by remembering $C$?"*. The personal context $C$ is open-ended rather than schema-bound. By requiring personalization grounded in declarative memories, *MIP-Bench* enables attribution (e.g., "this answer used C1 and C2"), controllability to edit $C$, and safety (inspect which memories were used).

**Personalization Evaluation.** In non-declarative memory grounded personalization (Shehmir & Kashef, 2025; Harper & Konstan, 2015; Ni et al., 2019; Salemi et al., 2023; Kumar et al., 2024), the justification for gold labels relies on the assumption that each response reflects the latent habits or patterns of a consistent user, and evaluation thus measures how well outputs reproduce user behavior.

Standard task-specific metrics have been applied, such as precision, recall, accuracy, F1, NDCG, and Hit Ratio for classification, MAE and RMSE for regression, and text similarity metrics including ROUGE and METEOR for generation. PersonalLLM (Zollo et al., 2024) instead leverages user-specific RLHF reward scores as evaluation signals, directly measuring alignment with preferences as captured by reward models. Explicit personalization benchmarks utilize metrics that tests alignment with the specified personal context: PersoBench (Afzoon et al., 2024) introduces Consistency Score and Persona Distance (P-Dist) to quantify personalization, while Personalized Soups (Jang et al., 2023) employs pairwise evaluation, where either GPT-4 or human judges determine which response better aligns with a given personal context. These evaluations operate at the instance level, assuming a single user for each query. However, personalization inherently requires considering effects across the user distribution—assessing not only how well a change benefits one user, but also how it impacts others. To address this, *MIP-Bench* introduces distribution-level metrics, IPS and DPS, providing a more holistic and balanced view of personalization performance.

## 3  THE *MIP-Bench* BENCHMARK

### 3.1  TASK FORMULATION

**Users and Histories.**  We have $N$ registered users $U = \{U_1, \ldots, U_N\}$. Each user $U_i$ comes with a dialog history $S_i = \big[(t_1, s_1), \ldots, (t_j, s_j)\big]$, where $s_j$ denotes a multi-turn conversation session that took place at time $t_j$.

**Test Episodes.**  At test time $t_q$, an identical question $Q$ is posed to $M \leq N$ users. Among those $M$ users, there $K$ *distinct preferences* that determine what an *ideal* answer to question $Q$ looks like. Each preference is grounded in some personal context $C_i \subseteq S_i$ of a user's dialog history.

**Rubrics.**  For every question $Q$, we provide a fixed set of $K$ binary rubrics $R = \{r_1, \ldots, r_K\}$. Each rubric is a simple yes-or-no probe such as *"Is malaria considered a likely diagnosis in the response?"* Rubrics convert a free-form answer into a $K$-bit vector.

**Ground-truth Preference.**  The $K$ distinct preferences are each represented as a one-hot binary vector of length $K$. For user $U_i$, the ground-truth preference is $\mathbf{y}_i = (y_{i1}, \ldots, y_{iK}) \in \{0,1\}^K$, where $\sum_{k=1}^{K} y_{ik} = 1$.

**Model Prediction.**  A model $f$ returns a free-form answer $a_i = f(U_i, Q)$. Applying the rubrics to $a_i$ yields the predicted preference, $\hat{\mathbf{y}}_i = (\hat{y}_{i1}, \ldots, \hat{y}_{iK})$.

### 3.2  EVALUATION PROTOCOL

Personalization should be judged both from the perspective of each individual user and across the population as a whole. We therefore introduce two complementary metrics.

**Instance Personalization Score (IPS).**  For a single question-user pair $(Q, U_i)$, we compare the one-hot ground-truth preference $\mathbf{y}_i$ with the rubric-derived prediction $\hat{\mathbf{y}}_i$ using the Jaccard index (Manning et al., 2008):

$$\text{IPS}(Q, U_i) = \frac{|\mathbf{y}_i \cap \hat{\mathbf{y}}_i|}{|\mathbf{y}_i \cup \hat{\mathbf{y}}_i|}. \tag{1}$$

Since $\mathbf{y}_i$ contains exactly one 1, IPS $= 1$ if the answer isolates the correct preference, $0 < \text{IPS} < 1$ if it mentions the correct preference *plus* extra, irrelevant ones, and IPS $= 0$ if it misses the correct preference entirely. A high IPS means that the answer hits the relevant preference instead of offering a catch-all response, signaling a stronger, more targeted personalization for the specific user. Hence, the average IPS reflects the degree of personalization that a typical user can expect when interacting with the model.

**Distribution Personalization Score (DPS).**  A high IPS alone does not guarantee that the model produces *different* answers for users with different needs. To explicitly reward correct differentia-

tion, we define

$$\text{DPS}(Q) = \frac{1}{M} \sum\nolimits_{i=1}^{M} \big[\text{IPS}(Q, U_i) \cdot w_i\big], \quad w_i = \frac{\left\|\hat{\mathbf{y}}_i - \bar{\mathbf{y}}\right\|_2^2}{\sum_{j=1}^{M}\left\|\hat{\mathbf{y}}_j - \bar{\mathbf{y}}\right\|_2^2}, \quad \bar{\mathbf{y}} = \frac{1}{M} \sum\nolimits_{i=1}^{M} \hat{\mathbf{y}}_i. \quad (2)$$

Here, $w_i$ is a normalized weight that represents the *relative deviation* of each predicted response $\hat{\mathbf{y}}_i$ from the population mean $\bar{\mathbf{y}}$, ensuring that $\sum_{i=1}^{M} w_i = 1$. Thus, DPS aggregates the correctness of model responses (IPS) weighted by their relative deviation ($w_i$), measuring the extent of *accurate differentiation*. Since $0 \leq \text{IPS} \leq 1$ and the weights are normalized, the resulting DPS is naturally bounded between 0 and 1, making it a stable measure of distribution-level personalization.

IPS measures individual-level personalization, indicating how well the model aligns with a specific user's preferences. However, it does not reveal whether the model outputs the same response for all users. DPS evaluates *differentiation*; when user preferences are heterogeneous, it assigns higher scores if the model produces multiple, distinct responses. While IPS quantifies user-level alignment, DPS together ensures the model as a whole avoids *one-answer-fits-all* behavior.

## 4 THE *MIP-Bench* DATASET

### 4.1 THE DATA SCHEMA

**Users.** *MIP-Bench* consists of 379 synthetic users $U = \{U_1, U_2, \ldots, U_N\}$.. Each user $U_i$ is associated with a timestamped interaction history with an AI assistant, $S_i = \big[(t_1, s_1), (t_2, s_2), \ldots, (t_j, s_j)\big]$. In total, the dataset contains 74,284 conversation sessions.

**Personalization Graphs and Personalization Paths.** *MIP-Bench* contains 514 Personalization Graphs with a total of 3,285 personalization paths. Of these graphs, 250 are from the casual domain and 264 from the expert domain, including 149 legal and 115 medical. Each personalization graph encodes the reasoning steps required to personalize a response. The root node is a common query $Q$—a question whose appropriate answer differs across a subset of users $U_Q \subseteq U$ due to personal context. Intermediate nodes are recollection queries that probe long-term memory. For each user $U_i \in U_Q$, a personal context $C_i \subseteq S_i$ supplies answers to these recollection queries. The leaf node is a ground-truth preference vector $\mathbf{y}_i \in \{0, 1\}^K$ over a rubric set $R = \{r_1, \ldots, r_K\}$, where rubrics are binary probes that capture decisive distinctions among possible answers. Thus, each personalization path from the root to a leaf specifies a unique, context-justified personalization trajectory for a user $U_i$.

### 4.2 THE DATA CONSTRUCTION PIPELINE

Central to the construction of *MIP-Bench* is the grouping of personalization paths into coherent user profiles. This grouping is guided by two key considerations. First, we account for the population distribution of response preferences, $\{\mathbf{y}_i\}_{i=1}^{M}$, to ensure that IPS captures the expected degree of personalization experienced by an average user. Second, we enforce compatibility among personal contexts, so that each user exhibits a consistent interaction history $S_i$, without contradictions that would undermine the rationale for personalization. By default, we employ GPT-4o (OpenAI, 2024) as the generation engine. Figure 3 presents the overall pipeline, with each stage described in detail below.

#### 4.2.1 STEP 1: MULTI-PATH QA GENERATION

We begin by collecting the common query $Q$, defined as a question whose appropriate answer varies depending on personal context. In the expert domain, we adapt 149 legal questions from ConditionalQA (Sun et al., 2021), 61 medical questions from DDXPlus (Fansi Tchango et al., 2022), and an additional 54 from MediQ (Li et al., 2024). In the casual domain, we draw 32 questions from PersonalLLM (Zollo et al., 2024), 6 from LongMemEval (Wu et al., 2024), 14 from Locomo (Maharana et al., 2024), 74 from MultiChallenge (Sirdeshmukh et al., 2025), and 124 from IP-Dialog (Peng et al., 2025).

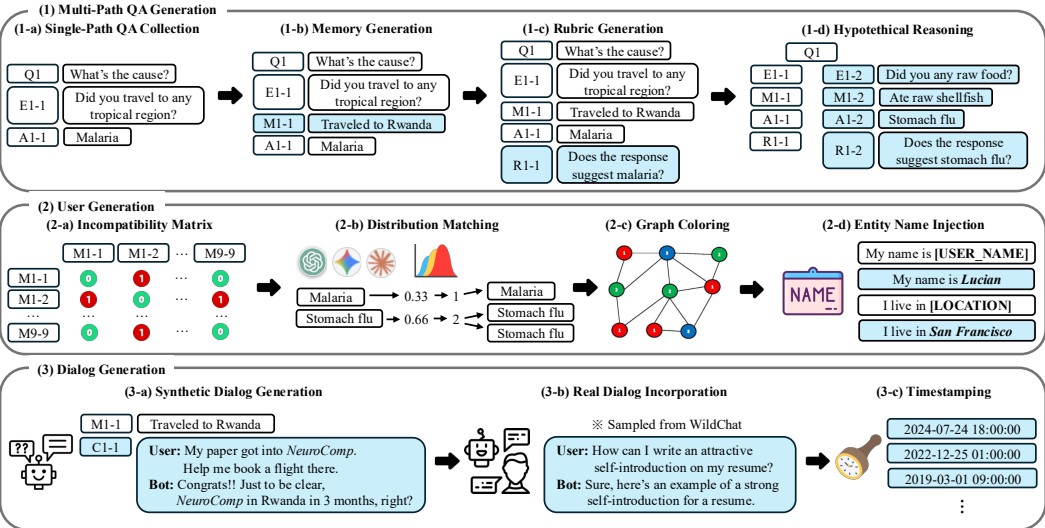

Figure 3: A detailed illustration of the data construction pipeline. Here, $Q$ represents the common query, $E$ a set of recollection queries, $M$ memory items, $A$ short answers, and $R$ rubrics. $C$ denotes the conversation session that provides the personal context.

Many of these queries are originally single-path: each has a short answer $A_{i-1}$ supported by a set of recollection queries $E_{i-1}$. To construct multi-path personalization graphs, we apply hypothetical reasoning: we alter the set of recollection queries to $E_{i-2}$ and verify whether this change yields a distinct short answer $A_{i-2}$. In the expert domain, all expanded personalization paths are validated by reviewers with domain expertise—law school students for legal questions and medically trained authors for medical questions. Each recollection query $E_{i-j}$ is then transformed into a memory item $M_{i-j}$, representing a concrete episode that supports the query. Likewise, short answers $A_{i-j}$ are converted into rubrics, which probe the most critical points of distinction among possible answers. This step results in 514 personalization graphs with 1,477 personalization paths.

### 4.2.2 STEP 2: USER GENERATION

We generate users by grouping personalization paths. To begin, we construct an incompatibility matrix across 1,088,068 distinct memory item pairs $(M_{i-j})$ using GPT-4o-mini (OpenAI, 2024). In parallel, we approximate the population distribution over short answers using aggregated log probabilities from multiple LLMs (OpenAI, 2024; Team et al., 2024; Anthropic, 2024). During this process, the same memory item $(M_{i-j})$ may be duplicated across multiple instances. The expanded incompatibility matrix is then processed with a graph-coloring approach to form user groups: vertices represent memory item instances, edges denote incompatibilities, and colors correspond to distinct users. We adapt the DSATUR heuristic (Brélaz, 1979), which scales effectively at our edge density ($\rho = 0.15$). The resulting clusters of memory items define each user $U_i$.

### 4.2.3 STEP 3: DIALOG GENERATION

We generate a timestamped conversation session $(t_j, s_j)$ for each memory item $M_{i-j}$, yielding a total of 3,285 synthesized sessions. To enrich user histories, we incorporate real human dialogs from WildChat (Zhao et al., 2024) This integration balances synthetic personal-context conversations with naturalistic interactions, thereby expanding the search space and ensuring a more even distribution of timestamps.

## 5 EXPERIMENTS

We conduct experiments to evaluate the effectiveness of *MIP-Bench* and analyze the personalization capabilities of recent large language models (LLMs). We aim to answer the following questions:

**Q1.** How do LLMs differ to perform implicit personalization?

**Q2.** Do the proposed IPS and DPS metrics effectively capture personalization quality?

**Q3.** What are the dominant failures of LLMs in implicit personalization?

**Q4.** What is the main bottleneck in implicit personalization: retrieval or generation?

**Q5.** How do different retrieval methods affect personalization quality?

**Experiment Setup.** We evaluate five LLMs, including three proprietary models of GPT-4o (OpenAI, 2024), Gemini-1.5-Pro (Team et al., 2024), and Claude-3.5-Sonnet (Anthropic, 2024), and two open-source models of Llama-3.1-70B and 8B (Dubey et al., 2024). We test all models on the same set of user profiles, queries, and histories in *MIP-Bench*. We measure Accuracy, IPS, and DPS values. Unless otherwise noted, all experiments use our proposed ersonalization graph pipeline. It is a graph-based retrieval method where each user query is expanded into a structured set of intents, which identify relevant segments of a user's profile or history before generation.

## 5.1 RESULTS

**Performance of State-of-the-Arts.** Table 1 compares overall performance across models. GPT-4o attains the highest accuracy, suggesting it is the best follower of general task instructions. However, Gemini-1.5-Pro outperforms all others on both IPS and DPS, indicating a superior ability to align responses with user-specific contexts. This reveals a key insight: *high accuracy does not guarantee effective personalization*. Personalized models must distinguish between users with similar queries but differing histories, which are only captured by IPS and DPS. Claude-3.5-Sonnet shows competitive IPS performance but lags in DPS, suggesting it may capture individual preferences but struggles to differentiate between users. Smaller models like Llama-3.1-8B fall short on all metrics, underscoring the need for both scale and context-aware mechanisms in effective personalization.

**Failure Mode Analysis.** We examine two major personalization failure modes: *majority guessing* and *catch-all responses*. Figure 4 shows that Claude-3.5-Sonnet frequently defaults to the most common answers regardless of user context, while GPT-4o often produces vague, multi-option responses that hedge rather than commit. These behaviors reduce personalization quality by collapsing response diversity, as reflected in lower DPS.

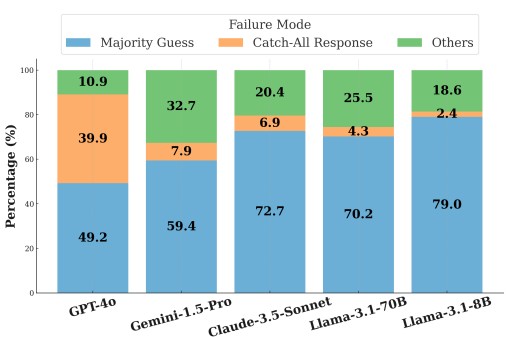

| Model | Acc. | IPS | DPS |
|---|---|---|---|
| GPT-4o | **0.845** | 0.642 | 0.315 |
| Gemini-1.5-Pro | 0.812 | **0.655** | **0.352** |
| Claude-3.5-Sonnet | 0.783 | 0.627 | 0.289 |
| Llama-3.1-70B | 0.725 | 0.556 | 0.225 |
| Llama-3.1-8B | 0.694 | 0.487 | 0.177 |

Table 1: Benchmark results on MIP-Bench.

Figure 4: Failure mode analysis. GPT-4o exhibits the most catch-all responses, while others frequently resort to majority guessing.

**Retrieval Bottleneck Analysis.** Table 2 highlights that when irrelevant histories (Random Context) are fed, performance drops sharply across all metrics. By contrast, access to ground-truth relevant context dramatically boosts both IPS and DPS. These results confirm that retrieval quality—rather than generation quality—is the primary bottleneck in personalization.

**Retrieval Design Analysis.** We test different retrieval designs. Table 3 compares simple *Long Context* prompting. *Personalization Graph* retrieval consistently outperforms other variants, especially in DPS. This demonstrates that effective personalization requires not only long memory, but

complex reasoning over which parts of a user's profile are relevant—an ability captured by graph-based retrieval.

| Retrieval Method | Acc. | IPS | DPS |
|---|---|---|---|
| Random Context | 0.652 | 0.204 | 0.065 |
| RAG (w/ Pers. Graph) | 0.845 | 0.642 | 0.315 |
| Ground-Truth Context | **0.947** | **0.921** | **0.601** |

Table 2: Retrieval quality comparison.

| Retrieval Method | Acc. | IPS | DPS |
|---|---|---|---|
| Long Context | 0.794 | 0.572 | 0.254 |
| RAG (w/o Pers. Graph) | 0.823 | 0.606 | 0.287 |
| RAG (w/ Pers. Graph) | **0.845** | **0.642** | **0.315** |

Table 3: Comparison of retrieval methods.

**Qualitative Analysis.** Examples reveal behavioral differences that accuracy alone cannot capture. In the legal domain, the query "Can I break my lease without penalty?" was posed to multiple users. One user's history revealed that they lived in California and had repeatedly complained about severe mold in their apartment, while another resided in New York and had previously discussed wanting to move closer to work. Gemini-1.5-Pro tailored its responses, citing California tenant laws that allow termination under uninhabitable conditions for the first user, and outlining standard lease obligations with potential negotiation strategies for the second. GPT-4o instead provided a generic checklist—"review your lease, talk to your landlord, consult a lawyer"—while Claude-3.5-Sonnet defaulted to a vague answer: "You usually cannot break a lease without paying a penalty."

In the casual domain, when asked "What should I cook for dinner tonight?", one user had previously expressed enthusiasm for vegetarian meals, while another had discussed following a keto diet. Gemini-1.5-Pro adapted accordingly, suggesting lentil curry for the vegetarian user and grilled salmon with avocado salad for the keto-focused user. GPT-4o produced a long mixed list including both vegetarian and non-vegetarian options, diluting personalization, and Claude-3.5-Sonnet ignored dietary history altogether, recommending pasta.

These examples demonstrate that personalization difficulty grows with the reasoning depth required for retrieval. Legal queries demand synthesizing personal details with jurisdiction-specific rules, while casual queries test whether models can recall and apply everyday preferences. Accuracy metrics alone cannot capture whether models collapse to one-size-fits-all behavior, but qualitative outputs reveal that differentiation across users remains the key challenge—precisely what IPS and DPS are designed to measure.

**Key Takeaways.** In summary, our experimental findings yield several insights:

- **Gemini-1.5-Pro** shows the strongest personalization performance, achieving the highest IPS and DPS by adapting responses precisely to individual users.

- **GPT-4o** attains the highest overall accuracy, but often overgeneralizes, generating catch-all responses that lower its DPS.

- **Claude-3.5-Sonnet** frequently resorts to majority guessing, returning the most common answer and failing to differentiate between users.

- **Llama models** underperform across all metrics, indicating challenges in integrating retrieved context and generating user-specific outputs.

- **Retrieval, not generation, is the primary bottleneck** for effective personalization. Simply increasing input length via long context prompting is insufficient.

- **Structured retrieval methods**, such as the *Personalization Graph*, significantly improve performance by supplying relevant and focused evidence.

## 6 CONCLUSION

We introduce *MIP-Bench*, designed to evaluate implicit personalization (Peng et al., 2025; Jin et al., 2024) in large language models. Unlike prior work, *MIP-Bench* challenges models to infer which parts of a user's history matter—without explicit personalization cues in prompts—and rewards meaningful variation across users through both instance-level and distribution-level metrics (IPS

and DPS). Our experiments find multiple insightful findings about recent LLMs' personalization capability. Furthermore, using *Personalization Graph* consistently improves the retrieval performance over long context baselines. *MIP-Bench* provides a unified evaluation setting for studying personalization (Zhang et al., 2024b) as it naturally arises in the real-world language assistant use. We hope this benchmark encourages future research into more context-sensitive, user-adaptive models that move beyond generic responses and better reflect individual users' needs.

## 7 LIMITATIONS

While *MIP-Bench* provides a novel framework for evaluating implicit personalization (Peng et al., 2025; Jin et al., 2024), it has several limitations. First, the benchmark focuses on text-based user histories and does not consider other modalities such as user behavior logs, images, or voice interactions, which could provide richer context in real-world systems. Second, although we simulate diverse user profiles and histories, the scale of *MIP-Bench* is smaller than that of production systems, potentially limiting generalizability. Third, personalization is measured using reference answers curated by annotators, which may not capture all valid user-specific responses—especially in subjective queries. Finally, while DPS rewards differentiation, they may overlook subtler personalization dimensions such as tone, style, or politeness unless these correlate with content-level changes. We view *MIP-Bench* as a first step toward principled evaluation of implicit personalization and hope future work extends its scope, fidelity, and scale.

## 8 REPRODUCIBILITY STATEMENT

We ensure reproducibility by providing an anonymized code containing the benchmark data and code (link in the abstract), which allows others to rerun our pipelines and metrics. The task formulation and evaluation protocol are described in Section 3, where the formal definitions of *IPS* (Eq. (1)) and *DPS* (Eq. (2)) provide exact scoring rules, and Appendix B outlines the mathematical properties and proof sketches of *DPS*. Details of dataset construction, including the end-to-end pipeline, personalization graphs, and timestamping logic, are provided in Section 4.2. Appendix C contains the full profile taxonomy and schema.

## 9 ETHICS STATEMENT

We acknowledge and adhere to the ICLR Code of Ethics. In constructing MIP-Bench, we ensured that no personally identifiable information was used and that all synthetic and incorporated dialog data were processed in ways that respect privacy and confidentiality. Our work focuses on advancing methods for evaluating memory-driven implicit personalization in large language models while explicitly safeguarding user privacy, thereby mitigating potential risks of misuse. By enabling more context-sensitive and adaptive personalization, our benchmark aims to support the development of AI systems that enhance social welfare through safer, more trustworthy, and user-aligned applications.

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

# A CASE STUDY: A SAMPLE EVALUATION ON FIGURE 1

Consider the example in Figure 1, where the same question is presented to $M = 3$ users, who exhibit $K = 2$ distinct preferences regarding what constitutes an ideal answer. Accordingly, we define $K = 2$ binary rubrics $R = \{r_1, r_2\}$, where $r_1$ asks *'Is malaria considered a likely diagnosis in the response?'*, and $r_2$ asks *'Is stomach flu considered a likely diagnosis in the response?'*.

The ground-truth (GT) preference for User 1 is $y_1 = (1, 0)$, grounded in his personal context $C_1$—a session noting recent travel to Rwanda, a malaria-endemic region. For User 2–3, the GT preferences are $y_2 = y_3 = (0, 1)$, grounded respectively in $C_2$, eating raw shellfish the previous day, and $C_3$, having close contact with someone who vomited. The model $f$ produces a free-form answer for each user: $a_1 = f(U_1, Q)$, $a_2 = f(U_2, Q)$, and $a_3 = f(U_3, Q)$. These responses are then evaluated against the binary rubrics $R$ to determine how well they align with each user's ground-truth preferences.

## A.1 CALCULATING IPS

We now illustrate how the Instance-level Personalization Score (IPS) is computed for User 1 across three different model responses.

**Case 1.** Suppose $a_1$ is a personalized response tailored to User 1's context:

*"The most likely cause to urgently rule out is malaria, which is common in Rwanda..."*

Applying rubric $r_1$ and $r_2$, we obtain the predicted preference $\hat{y}_1 = (1, 0)$, indicating the response zeros in on malaria. Since the ground-truth is $y_1 = (1, 0)$, we compute:

$$\text{IPS}(Q, U_1) = \text{Jaccard}((1, 0), (1, 0)) = \frac{1}{1} = 1.$$

**Case 2.** Now consider a generic response that lists multiple possible causes:

*"Some possibilities include stomach flu (viral gastroenteritis), food poisoning, malaria, dengue fever..."*

Applying the rubrics yields $\hat{y}_1 = (1, 1)$, meaning the response includes both malaria and stomach flu. Comparing with the ground-truth:

$$\text{IPS}(Q, U_1) = \text{Jaccard}((1, 0), (1, 1)) = \frac{1}{2} = 0.5.$$

**Case 3.** Finally, consider a case where the model collapses to the majority preference, ignoring User 1's personal context:

*"Sounds like stomach flu..."*

This yields $\hat{y}_1 = (0, 1)$. Since the ground-truth is $(1, 0)$, we get:

$$\text{IPS}(Q, U_1) = \text{Jaccard}((1, 0), (0, 1)) = \frac{0}{2} = 0.$$

IPS decreases from Case 1 to Case 3, aligning with the progressively weaker degree of personalization to each user's personal context.

## A.2 CALCULATING DPS

We now illustrate how the Distribution-level Personalization Score (DPS) is computed for $Q$ across three scenarios, each with a different set of predicted preferences $\{\hat{y}_1, \hat{y}_2, \hat{y}_3\}$.

**Case 1.** Suppose the model produces perfectly personalized responses $a_1, a_2, a_3$ for all users. Applying the rubrics, we obtain the predicted preferences $\hat{y}_1, \hat{y}_2, \hat{y}_3$ and the mean preference $\bar{y}$:

$$\hat{y}_1 = y_1 = (1, 0), \quad \hat{y}_2 = y_2 = (0, 1), \quad \hat{y}_3 = y_3 = (0, 1).$$

$$\bar{\mathbf{y}} = (1/3, 2/3).$$

Since the predicted preferences match the ground-truth preferences exactly, we have

$$IPS(Q, U_i) = 1 \quad \text{for all } i \in \{1, 2, 3\}.$$

and

$$\text{DPS}(Q) = \frac{1}{3}\Big[ 1 \cdot \|(1,0) - (1/3, 2/3)\|_2^2$$
$$+ 1 \cdot \|(0,1) - (1/3, 2/3)\|_2^2 + 1 \cdot \|(0,1) - (1/3, 2/3)\|_2^2 \Big]$$
$$= \frac{1}{3}\left[ \frac{8}{9} + \frac{2}{9} + \frac{2}{9} \right] = \frac{4}{9} \approx 0.444$$

**Case 2.** Now suppose the model collapses to a single majority preference, disregarding user-specific contexts:

$$\hat{y}_1 = \hat{y}_2 = \hat{y}_3 = (0,1). \qquad \bar{\mathbf{y}} = (0,1)$$

In this case, each predicted preference $\hat{\mathbf{y}}_i$ is identical to the mean $\bar{\mathbf{y}}$, so the squared deviation term $\|\hat{\mathbf{y}}_i - \bar{\mathbf{y}}\|_2^2$ becomes zero for all $i$. Therefore,

$$\text{DPS}(Q) = 0.$$

Note that DPS remains zero regardless of the specific preference the model defaults to. For instance, even if $\hat{y}_1 = \hat{y}_2 = \hat{y}_3 = (1,1)$, the lack of differentiation still results in a DPS of zero.

**Case 3.** Finally, consider a model that attempts differentiation but mispersonalizes one user - User 2:

$$\hat{y}_1 = \hat{y}_2 = (1,0), \quad \hat{y}_3 = (0,1). \qquad \bar{\mathbf{y}} = (2/3, 1/3)$$

$$\text{DPS}(Q) = \frac{1}{3}\left[ 1 \cdot \frac{2}{9} + 0 \cdot \frac{2}{9} + 1 \cdot \frac{8}{9} \right] = \frac{10}{27} \approx 0.37$$

Although the average IPS in Case 3 is identical to that in Case 2, the predicted responses in Case 3 exhibit greater variance. From a distributional standpoint, Case 3 represents a meaningful attempt to personalize, despite imperfections. DPS captures this distinction by assigning a higher score, reflecting the model's effort to differentiate across users.

In summary, while IPS measures user-level alignment, DPS quantifies whether the model meaningfully diversifies its outputs across a population, thereby offering complementary perspectives on personalization quality.

## B  MATHEMATICAL PROOF OF DPS

**Proposition (Optimality of perfect predictions).**  Let $p_k = \frac{1}{M}\sum_{i=1}^{M} \mathbf{1}[y_{ik} = 1]$ be the empirical ground-truth proportion for preference $k$ ($k = 1, \ldots, K$). For any set of predictions $\{\hat{\mathbf{y}}_i\}_{i=1}^{M}$ the Distribution-level Personalisation Score satisfies

$$\text{DPS}(Q) \leq 1 - \sum_{k=1}^{K} p_k^2, \tag{1}$$

and equality in equation 1 holds iff $\hat{\mathbf{y}}_i = \mathbf{y}_i$ (hence IPS $= 1$) for every user $i$.

Write $w_i = \text{IPS}(Q, U_i) \in [0,1]$ and $d_i = \|\hat{\mathbf{y}}_i - \bar{\mathbf{y}}\|_2^2$. Then, by definition, $\text{DPS}(Q) = \frac{1}{M}\sum_{i=1}^{M} w_i\, d_i \leq \frac{1}{M}\sum_{i=1}^{M} d_i$ because $w_i \leq 1$. Hence

$$\text{DPS}(Q) \leq \frac{1}{M}\sum_{i=1}^{M} \|\hat{\mathbf{y}}_i - \bar{\mathbf{y}}\|_2^2 = \frac{1}{M}\sum_{i=1}^{M} \|\hat{\mathbf{y}}_i\|_2^2 - \|\bar{\mathbf{y}}\|_2^2. \tag{2}$$

*Step 1: Tightest bound on the right-hand side.* The term $\|\hat{\mathbf{y}}_i\|_2^2$ equals the number of ones in $\hat{\mathbf{y}}_i$. Because $w_i = 1$ only when the prediction is the *single* correct label, maximising the right-hand side of equation 1 forces $\|\hat{\mathbf{y}}_i\|_2^2 = 1$ for every $i$ and, in turn, $w_i = 1$. With exactly one 1 per vector, $\sum_i \|\hat{\mathbf{y}}_i\|_2^2 = M$. Putting this into equation 1 gives

$$\mathrm{DPS}(Q) \ \leq \ 1 - \|\bar{\mathbf{y}}\|_2^2. \tag{3}$$

*Step 2: Expressing $\bar{\mathbf{y}}$ through $p_k$.* Under the one-hot constraint, $\bar{\mathbf{y}} = (p_1, \ldots, p_K)$, so $\|\bar{\mathbf{y}}\|_2^2 = \sum_k p_k^2$. Equation equation 1 follows.

*Step 3: When is the bound attained?* Equation equation 1 becomes an equality iff both (i) $w_i = 1$ for every $i$ (otherwise the first inequality is strict) and (ii) each $w_i = 1$ prediction still carries a single 1 (otherwise we would violate the one-hot count used in Step 1). These two requirements together force $\hat{\mathbf{y}}_i = \mathbf{y}_i \ \forall i$.

**Corollary.** The maximal achievable DPS is $\mathrm{DPS}_{\max} = 1 - \sum_k p_k^2$. It is strictly smaller than 1 whenever the preference distribution is imbalanced.

## C  PROFILE TAXONOMY

To support diverse and realistic personalization scenarios, *MIP-Bench* defines a structured taxonomy of user profile attributes. This taxonomy ensures both coverage and control: it captures a wide range of possible user traits while enabling consistent annotation and evaluation across users and queries.

Each user profile consists of values drawn from multiple categories (e.g., *location*, *profession*, *health*, *travel history*), which are subdivided into well-defined subcategories. These subcategories are labeled with usage semantics to guide query generation and retrieval relevance.

Specifically, we introduce two key design indicators:

- **Single Active** (**O**): only one value in this subcategory is considered active at a time (e.g., current city of residence). Values with the $\triangle$ symbol are *typically* single but can occasionally have multiple active values (e.g., *symptoms*).
- **Named Entity** (**O**): values may contain synthetic named entities such as people, organizations, or places. These fields are useful for testing name-based disambiguation and retrieval.

We refer to this structured design as a *profile schema*, which provides the backbone for query construction, retrieval candidate selection, and ground-truth annotation.

Table 4 presents the full taxonomy used in *MIP-Bench*. It includes 24 subcategories spanning demographics, health, travel, education, and more. By organizing user traits in this way, we enable fine-grained control over personalization cues and ensure systematic coverage of plausible query contexts.

This taxonomy also allows for scalable expansion and domain-specific filtering, making *MIP-Bench* adaptable to a range of downstream personalization tasks and evaluation settings.

| Category | Sub Category | Single Active | Named Entity |
|---|---|---|---|
| name | name | O | O |
| age | age | O | X |
| gender | sex_assigned_at_birth | O | X |
| | gender_identity | O | X |
| location | birthplace | O | X |
| | residence | O | X |
| | nationality | △ | X |
| employment | job_status | O | X |
| | profession | △ | X |
| | workplace | △ | O |
| school | school_status | O | X |
| | school | △ | O |
| | degree | △ | X |
| | degree_subject | X | X |
| possession | animal | X | O |
| | vehicle | X | X |
| | license | X | X |
| | financial_asset | X | X |
| | real_estate | X | X |
| | other_asset | X | X |
| language | language | X | X |
| social_relation | marital_status | O | X |
| | family_status | O | X |
| | relationship | X | O |
| physical_attribute | physical_attribute | X | X |
| personality_trait | personality_trait | X | X |
| preference | food | X | X |
| | drink | X | X |
| | movie | X | O |
| | media | X | O |
| | book | X | O |
| | music | X | O |
| | sport | X | X |
| | activity | X | X |
| | location | X | O |
| | person | X | O |
| | organization | X | O |
| | color | X | X |
| | season | X | X |
| | animal | X | X |
| | sexual_orientation | X | X |
| | political_orientation | X | X |
| | religion | X | X |
| medical_history | physical | X | X |
| | psychiatric | X | X |

Table 4: Taxonomy of profile categories.

