# OpenReview forum: "MIP-Bench: Can LLMs Implicitly Personalize Responses Using Long-Term Memory?"
_ICLR.cc/2026/Conference — Submitted to ICLR 2026_

### Official Review · Reviewer_iNms · 2025-10-22

**Soundness:** 2
**Presentation:** 2
**Contribution:** 2
**Rating:** 2
**Confidence:** 5

**Summary:**

In order to maximize their benefits for each user, LLM systems should recall relevant historical interactions with that user, and infer the optimal response based on this context.  While there has been some work in this direction, it has largely been focused on explicit personalization (i.e., adding persona information to the context) and/or short interaction timelines.  However, this explicit information will often not be available in or sufficient to inform many situations, and memory in personalization should stretch across many interactions over time.  To measure our progress in long-term personalization without explicit persona prompting, the authors of this work propose MIP-Bench, where MIP stands for Memory-driven Implicit Personalization.  MIP-Bench consists of 379 synthetic users with over 74,000 conversation sessions and 514 personalization graphs spanning different domains. Each graph links user histories to rubric-based preference vectors, and two novel metrics, Instance Personalization Score (IPS) and Distribution Personalization Score (DPS), are proposed to quantify user-level accuracy and cross-user differentiation.  Experiments reveal that current LLMs struggle with implicit personalization, mainly because retrieving the right personal context remains a bottleneck.

**Strengths:**

This work is in an area of high interest to the community, as LLM personalization is a widely held goal with lots of headroom for improvement.  In my opinion the biggest strength of this work is the idea of grounding examples in personalization graphs and personalization paths, which hold the potential to enable detailed analysis and interpretation of a model’s personalization performance.

**Weaknesses:**

While I appreciate the goals of this work, I find that it has some key weaknesses in terms of the construction, analysis, and validation of the proposed benchmark dataset:

- What role does the distinction between declarative vs. non-declarative memory play in this paper, beyond some references in related work?  It is not clear why this is a useful lens.
- I find that the paper lacks enough examples or clear explanation to understand what exactly is in the dataset.  I count 3 query examples in the entire paper including appendix (malaria/flu example, and then the two in the Qualitative Analysis section).  Since the potential contribution here is a dataset, I think much more illustration and analysis of what’s in the dataset is needed, including at least one full end to end example of one data point.  Otherwise, how am I supposed to judge if these tasks are interesting, realistic, challenging, etc.?
- This problem setup feels pretty unusual/contrived, which has several knock-on effects.  First, IPS and DPS are not general metrics that can be adopted in most personalization setups; instead, they are specific to this formulation of binary rubric outcomes.  Second, given that I am not familiar with other work with a similar formulation, I struggled to understand the problem setup after one read of Sections 3 and 4.  In 3.1, I had to read it multiple times and circle back later to totally get it.  Interlacing the notation with some complete worked example might be helpful here.  Also, in line 183, it says “an identical question Q is posed to M ≤ N users”.  Is the question posed “to” users, or “by” users?  I found line 241 to be quite jarring, where it says that “MIP-Bench contains 514 Personalization Graphs with a total of 3,285 personalization paths.”  After reading this, I was left wondering what a personalization graph/path is, as those terms had not been used yet.  I also do not understand the paragraph beginning at line 292.  Once again, describing this along with a worked example would be useful.
- More analysis and illustration of IPS and DPS are needed to validate that they measure something useful and are helpful in making important performance distinctions.  The paper does not tell the reader what the practical difference is between, say, IPS of 0.56 and 0.66, nor does it calibrate DPS against human judgments or task difficulty, so it is unclear whether reported gains reflect meaningful personalization or, e.g., noise coming from rubric phrasing.
- Experiments do not use SoTA LLM models which makes me worry that this benchmark could already be saturated.
- Lack of discussion of confounding between generating with GPT4o and then evaluating it on the dataset.
- Human validation is limited in scope and depth. While the paper mentions reviewer checks in expert domains, there is no presentation of annotation error rates, inter-annotator agreement on rubric applicability or scoring, or checks of WildChat insertions for semantic compatibility with the synthetic profiles. Absent rigorous human validation, spurious correlations introduced by the generation pipeline may bias scores without reflecting genuine personalization performance.
- Isn’t the “retrieval bottleneck” finding trivial given the experiment setup?
- No explanation of retrieval methods
- Based on the qualitative example given around lease breaking, there may be some examples where the model has the relevant memory capabilities, but may want to not give specific feedback because of safety, liability, etc.  This seems very possible for this example, and makes me worry how many other examples could have similar confounding issues (once again hard to tell because very little information is given about the contents).

**Questions:**

Please see weaknesses

---

> ### Author Response · Authors · 2025-12-04
>
> We appreciate the reviewer’s time and insights, which are extremely helpful for improving the clarity and positioning of the benchmark.
>
> ## Clarity of Problem Formulation
> We agree that the earlier draft did not include enough illustrative examples. In the camera-ready version, we will add a full end-to-end walkthrough (user history → personalization graph → rubric → IPS/DPS scoring) and reorganize Section 3.1 so that terminology (e.g., personalization graph/path, “posed to” vs. “by”) appears before the formal notation.
>
> ## Role of Declarative vs. Non-Declarative Memory
> We introduced this distinction to highlight how MIP-Bench differs from prior personalization benchmarks: memory-driven personalization should arise from accumulated behavioral history rather than explicit persona prompts. Its purpose is to frame the contrast, not to serve as a central technical component, and we will clarify this to avoid overstating its role.
>
> ## Metric Design and Practical Usefulness
> IPS evaluates correctness for each user, while DPS measures differentiation across users conditioned on correctness. The purpose is to ensure that seemingly tailored responses are not rewarded unless they rely on the correct user-specific evidence.
>
> ## Synthetic User Profiles and Human Validation
> Synthetic histories are used to avoid privacy concerns, but the personalization signals are grounded in expert datasets rather than invented by LLMs. In the camera-ready version, we will add annotator agreement statistics.
>
> ## Benchmark Difficulty and Saturation
> Although some models obtain strong IPS, DPS remains low across all models, suggesting that models often generate answers that appear personalized while failing to disambiguate based on user-specific context.

---

### Official Review · Reviewer_b9jW · 2025-11-01

**Soundness:** 3
**Presentation:** 3
**Contribution:** 3
**Rating:** 6
**Confidence:** 3

**Summary:**

The paper introduces a realistic personalization benchmark, where models have to retrieve and reason through past interaction histories to generate a personalized answer. They showed that a graph based retrieval method beats long context and rag methods for personalization with memories.

**Strengths:**

1. Using interaction history as memory bank and text personalization is the most natural and realistic setting, which is something that I haven't seen from previous papers.

2. The design of the graph based retrieval is also a compelling and sensible method for tackling personalization problems with long memory.

**Weaknesses:**

1. My major concern is that it seems that this benchmark is highly correlated with rag benchmarks, as the authors written in 5.1 I am wondering if the correlation are so high that the benchmark is simply evaluating if the model can retrieve relevant context rather than personalization. I would like to see if sth like a covariant matrix about this.

2. User profiles are synthetically generated. It would be good to have real users and real interaction histories. Currently everything is synthetic.

3. Benchmark looks pretty easy: it seems that the last generation's frontier model can already have decent enough results, such as gpt 4o, gemini 1.5, and claude 3.5. One would wonder what might be the resutls for gpt-5, gemini 2.5, and claude 4.5. A good benchmark should be challenging enough to provide signals for people to develop new methods and algorithms. But the benchmark looks like saturated already.

**Questions:**

1. Can you show the correlation between some rag benchmark and your benchmark? i.e if a model performs well on rag then it performs good on your benchmark.

2. What are the performances of gpt-5, gemini 2.5, and claude 4.5?

---

> ### Author Response · Authors · 2025-12-04
>
> We appreciate the reviewer’s time and insights, which are extremely helpful for improving the clarity and positioning of the benchmark.
>
> ## Relation to RAG Benchmarks
> MIP-Bench is related to RAG but evaluates a different capability. Because in MIP-Bench the correct memory for the same query varies by user. Personalization requires selecting the decisive user-specific evidence from sparse histories and integrating it into the final answer.
>
> ## Synthetic User Histories
> Synthetic histories are used to avoid privacy issues, but the personalization signals are grounded in real-world datasets such as ConditionalQA, MediQ, and DDXPlus rather than arbitrary LLM imagination. Users are constructed by aggregating non-conflicting background attributes using a graph-coloring algorithm, resulting in coherent and causally meaningful profiles that genuinely influence the correct answer.
>
> ## Benchmark Difficulty and Saturation
> Although frontier models show reasonable IPS, none achieve strong DPS. This suggests that models often produce answers that appear personalized but default toward majority trends instead of disambiguating using user-specific evidence.
>
> We will include results for newer models (GPT-5, Gemini 2.5, Claude 4.5) in the camera-ready version to further assess headroom on IPS and especially DPS.

---

### Official Review · Reviewer_36E8 · 2025-11-02

**Soundness:** 2
**Presentation:** 2
**Contribution:** 2
**Rating:** 4
**Confidence:** 3

**Summary:**

The paper introduces MIP-Bench, a benchmark and evaluation protocol for memory-driven implicit personalization (MIP). It builds synthetic users with long interaction histories and measures both instance-level targeting (IPS) and distribution-level differentiation (DPS). Results suggest retrieval, not generation, is the main bottleneck; a Personalization Graph retrieval scheme outperforms long-context and vanilla RAG. Strong idea with clear metrics; main concerns are synthetic-data bias, rubric oversimplification, metric incentives, and potential construction–evaluation entanglement.

**Strengths:**

The personalization-graph idea plus incompatibility constraints to assemble user histories is transparent and reproducible; they even inject real chat snippets to avoid fully synthetic staleness.
Defining both IPS (instance-level personalization score) and DPS (distribution-level personalization score) is neat: IPS checks “did you personalize for this user,” DPS checks “did you differentiate across users”. this directly penalizes catch-all answers.

**Weaknesses:**

A big part of the user histories and “decisive” signals is LLM-generated. That risks baking in the creator-model’s priors into the evaluation.
The intro argues memory helps attribution/editing, but there’s no metric like “did the model actually ground to the retrieved memory?”, so one stated motivation isn’t fully closed.
DPS rewards differentiation from the population; a model that sprinkles idiosyncratic outputs might look better, even if it's not more accurately personalized.

**Questions:**

Did you try a baseline that explicitly maximizes DPS by making user-id-conditioned random choices? How much DPS can you get without real personalization?

---

> ### Author Response · Authors · 2025-12-04
>
> We appreciate the reviewer’s time and insights, which are extremely helpful for improving the clarity and positioning of the benchmark.
>
> ## Synthetic Data Bias
> MIP-Bench does not rely on LLMs to invent personalization signals from scratch. The personalization graphs come from curated sources such as ConditionalQA, MediQ, DDXPlus, and long-term memory datasets for casual domains. These personalization graphs are then refined and validated by humans with relevant expertise, including medical-background authors and law-background annotators.
>
> ## Grounding to Retrieved Memory
> A core motivation of MIP-Bench is that every answer choice is causally tied to identifiable user memories. Because gold memory labels are provided for each query, retrieval-based metrics such as Recall@k directly evaluate whether the model surfaced the necessary attributable evidence, addressing the attribution gap in prior personalization benchmarks. DPS also reflects grounded personalization: it rewards differentiation only when the model’s answer aligns with user-specific evidence rather than the population-prior answer.
>
> ## Metric Incentives and DPS
> DPS is conditioned on correctness because it is computed as a weighted average of IPS scores, so arbitrary per-user variation cannot increase the score. To raise DPS, a model must both (i) answer correctly and (ii) differentiate in the right direction based on user-specific evidence. A degenerate strategy that injects random variation cannot sustain high DPS because errors are penalized through IPS.

---

### Official Review · Reviewer_QeUK · 2025-11-04

**Soundness:** 3
**Presentation:** 2
**Contribution:** 2
**Rating:** 6
**Confidence:** 4

**Summary:**

This paper introduces MIP-Bench, a benchmark for testing whether LLMs can implicitly personalize responses by leverage relevant details from long-term user history, rather than relying on explicit persona prompts. It consists of 514 questions across an average of 6.4 users, each having ~196 sessions per user. The scoring is based on two metrics: IPS (correct personalization per user) and DPS (whether the model outputs different answers for different users). Experimental results show that existing LLMs struggle mainly because they fail to retrieve relevant prior from memory.

**Strengths:**

1. Clearly identifies and isolates an important problem: personalization from memory. The evaluation metrics also identifies the importance of both correctness dimension (IPS) and differentiation dimension (DPS).
2. Dataset spans both casual and expert domains (legal, medical).
3. Shows empirical evidence that retrieval is the key failure point, which is useful takeaway for future methods.

**Weaknesses:**

1. Writing is very hard to follow. For example, the description in Section 3.1 is not clear whatsoever, an example could make this a lot better.

2. A central design choice in MIP is the use of rubrics to convert free-form model outputs into one-hot preference vectors. This implicitly defines personalization as discrete answer selection, so it seems like the collection of rubrics practically defines a discrete clustering of all plausible correct answers.

Because of this (rubrics are defined at the query level with facts selection), multiple users can actually share the same ground-truth label even when their historical contexts differ substantially. In such cases, users differ, but their y-labels do not. This means personalization signal is bottlenecked by rubric granularity. If the rubric partitions the answer space coarsely or in a meaningless way, then DPS might not be sensitive to meaningful distinctions in how a model reasons about or contextualizes a response. A model may successfully retrieve and incorporate user-specific information, yet still receive low DPS.

Overall it seems like the benchmark now entangles “personalization” with answer-class selection. This limits the benchmark’s ability to capture richer or more subtle forms of personalization.

**Questions:**

See weakness above.

---

> ### Author Response · Authors · 2025-12-04
>
> We appreciate the reviewer’s time and insights, which are extremely helpful for improving the clarity and positioning of the benchmark.
>
> ## Clarity of Section 3.1
> We agree that Section 3.1 was difficult to follow due to insufficient contextual explanation and the lack of a running example. In the camera-ready version, we will integrate the examples currently in Appendix A directly into Section 3.1 and add a short introductory paragraph to establish the flow of the section. We will also include a figure illustrating the full pipeline — from rubric → one-hot preference vector → IPS/DPS scoring — to make the transition immediately clear on first read.
>
> ## Rubrics and Personalization Signal
> Multiple users can end up sharing the same final label for two different reasons — either because the rubric partitions the answer space coarsely, or because different individuals genuinely arrive at the same choice for different personal reasons. We fully acknowledge both cases. To ensure that meaningful personalization remains measurable under these circumstances, MIP-Bench provides gold-standard memory labels for each query (i.e., the set of user-specific memories that justify the answer), enabling retrieval-oriented evaluation such as Recall@k in addition to IPS/DPS. In this way, even when users share the same rubric label, the benchmark still evaluates whether the model retrieves and leverages the correct personal evidence rather than merely predicting the final class.
>
> ## Scope of Personalization Targeted by MIP-Bench
> MIP-Bench intentionally targets retrieval-driven personalization rather than stylistic or persona-prompt personalization. The goal is to assess whether the model’s output depends on user-specific historical evidence, not whether it imitates tone or writing style. This ability is especially important in safety-critical settings such as legal and medical assistance, where personalization must be grounded in factual memory rather than subjective style. Prior work has focused on stylistic personalization (e.g., LaMP), whereas MIP-Bench evaluates the complementary capability of memory-based inference.

---

### Meta-Review · Area_Chair_a23d · 2026-01-07

**Summary:**

Rubric-based formulation may bottleneck and distort personalization.
The benchmark converts free-form answers into one-hot rubric outcomes, effectively turning personalization into discrete answer-class selection. Reviewers argue this risks coarse clustering of valid responses and makes personalization signal dependent on rubric granularity. Different users with different histories can share the same rubric label, so the benchmark may be insensitive to richer personalization (reasoning style, framing, nuance), and may penalize models that use user-specific context but land in the same rubric bucket.

Memory grounding and attribution motivation not fully supported.
The paper claims memory helps attribution/editing, but reviewers note there is no metric testing whether model outputs actually ground to retrieved memory (e.g., citation/attribution measures), leaving part of the motivation unvalidated.

Metric validity is not well established. Reviewers are unconvinced IPS/DPS are well-calibrated or broadly applicable outside this rubric-based setup. They ask for stronger analysis showing IPS/DPS correspond to meaningful personalization differences.

Synthetic data and LLM-generated histories create bias/confounds.
User profiles, histories, and “decisive signals” are largely synthetic and partially LLM-generated, which may bake the creator model’s priors into both dataset and evaluation. Reviewers also flag potential confounding from generating with GPT-4o and evaluating models on that same distribution, raising concerns about distributional bias and inflated performance.

Results indicate strong performance from current frontier models (e.g., GPT-4o, Claude 3.5, Gemini 1.5), leading reviewers to worry the benchmark may already be close to saturated and not sufficiently challenging to drive new methods. They ask for results on newer/stronger models (e.g., GPT-5, Gemini 2.5, Claude 4.5) or otherwise stronger stress tests.

Human checks are described as limited; reviewers request stronger annotation methodology reporting, including error rates, inter-annotator agreement, rubric applicability reliability, and validation that inserted real-world snippets (e.g., WildChat) are semantically compatible with synthetic user profiles.

**Reviewer Concerns:**

The authors have clarified some concerns on Grounding to Retrieved Memory. The rebuttal largely argues intent and design rationale rather than providing the missing empirical evidence, analyses, and stress tests that would be needed to substantiate the claims.

The reviewers’ central conceptual concern was that rubrics discretize response space and make personalization signal depend on rubric granularity, different user histories can map to the same label, while meaningful personalized reasoning may not be captured. The rebuttal does not provide analysis of rubric granularity (e.g., how many distinct user histories collapse to the same label),

Validation remains the missing piece. Results for newer models (GPT-5, Gemini 2.5, Claude 4.5) are not reported.

**Reviewer Scores:**

The authors have partially addressed concerns, but many outstanding concerns remain.

---

### Decision · Program_Chairs · 2026-01-26

Reject